# Oceanic Anoxic Event 2 triggered by Kerguelen volcanism

C. A. Walker-Trivett [1,2] ✉, S. Kender [1,2] ✉, K. A. Bogus[1], K. Littler [1], T. Edvardsen[1], M. J. Leng[2], J. Lacey [2], J. B. Riding[2], I. L. Millar [2] & D. Wagner[2]

Large Igneous Provinces (LIPs) are associated with global warming and carbon cycle perturbations during Oceanic Anoxic Event 2 (OAE2, ~94 Ma) and the Mid-Cenomanian Event (MCE, ~96.5 Ma). However, there is still no consensus on the role volcanism played as a trigger, or its source – previously ascribed to the Caribbean LIP or High Arctic LIP. Here, we use Mentelle Basin sedimentary mercury (Hg) concentrations to determine the timing of volcanism, and neodymium (Nd) and strontium (Sr) isotopes for sedimentary provenance. High Hg concentrations compared to Northern Hemisphere records, and a shift to radiogenic Nd isotopes, indicates Kerguelen LIP volcanic activity and plateau uplift occurred in the lead up to and within OAE2. Whilst we find limited evidence that a volcanic event caused the MCE, pulsed Hg spikes before and during OAE2 imply volcanic emissions were key in driving climate and carbon cycle changes and triggering OAE2.

Oceanic Anoxic Event 2 (OAE2) occurred at the Cenomanian/Turonian boundary ( ~ 94 Ma) in the mid-Cretaceous during super-hot-house conditions, characterised by high atmospheric $CO_2$ and exceptionally high sea levels[1]. It was one of several periods of widespread ocean anoxia during the Mesozoic[2], and was coupled with deposition of organic-rich shales, marine extinctions[2,3] and terrestrial vegetation changes[4]. It is often expressed in the sedimentary record as a short-lived negative $\delta^{13}C$ carbon isotope excursion (CIE)[5] followed by a broad positive CIE in bulk organic matter ( ~ 5‰) and carbonate (2–3‰), although organic carbon isotopes do not always mirror carbonate trends[6]. The Mid-Cenomanian Event (MCE) is a smaller-scale, short-term anoxic event that occurred ~96.5 Ma, with a positive $\delta^{13}C$ excursion ~1‰[7,8], and a shift in foraminiferal, radiolarian, and calcareous nannofossil assemblages[9,10]. Initiation of OAE2 warming is thought to have enhanced nutrient cycling and stimulated high productivity, explaining the deposition of organic-rich material, the broad positive CIE, and the expansion of oxygen minimum zones (OMZs) due to enhanced decomposition within the water column[11]. This mechanism may also apply to the MCE but the event has hitherto been less comprehensively studied.

One mechanism proposed to have initiated OAE climate change is large-scale volcanism. During the Cretaceous there were several active Large Igneous Provinces (LIPs), including the Caribbean (CLIP;

~95–83 Ma and ~81–71 Ma), High Arctic (HALIP; ~130–90 Ma), and Kerguelen ( ~ 122–90 Ma) (Fig. 1). Each of these LIPs have major eruptive phase timings compatible with OAE2 and MCE initiation, although precise correlation between volcanism and these perturbations is under debate[12–14]. The evidence traditionally used to infer volcanism during OAE2 is sedimentary shifts in osmium (Os) isotopes, with many sections worldwide found to exhibit a characteristic negative $Os_i$ ($^{187}Os/^{188}Os$) shift from radiogenic (continental weathering) to unradiogenic (hydrothermal alteration and/or weathering of juvenile crust) values, and an increase in sedimentary Os concentrations[6,11,15–17]. Over OAE2, negative $Os_i$ shifts are variable between sites, and sometimes occurred in multiple pulses immediately prior to and within the event in the Western Interior Seaway (WIS), (proto-)North Atlantic Ocean, Tethys Ocean, European Epicontinental Sea, (proto-)South Atlantic Ocean, and the Pacific Ocean[6,11,16,18]. Identifying the precise OAE2 onset is critical to the debate but remains challenging in many records, particularly those without both carbonate and organic carbon isotope stratigraphy. High concentrations of Os in the WIS[18] and Canadian High Arctic[19] occur shortly after the initiation of a major phase of HALIP volcanism dated to ~97–93 Ma[14,20], potentially linking it with OAE2[18]. Other studies suggest that the CLIP was the trigger for OAE2[21,22], based on higher Os concentrations and large initial $Os_i$ excursions in the

[1]Camborne School of Mines, Department of Earth and Environmental Sciences, University of Exeter, Penryn Campus, Penryn TR10 9FE, UK. [2]British Geological Survey, Keyworth, Nottingham NG12 5GG, UK. ✉e-mail: cw725@exeter.ac.uk; s.kender@exeter.ac.uk

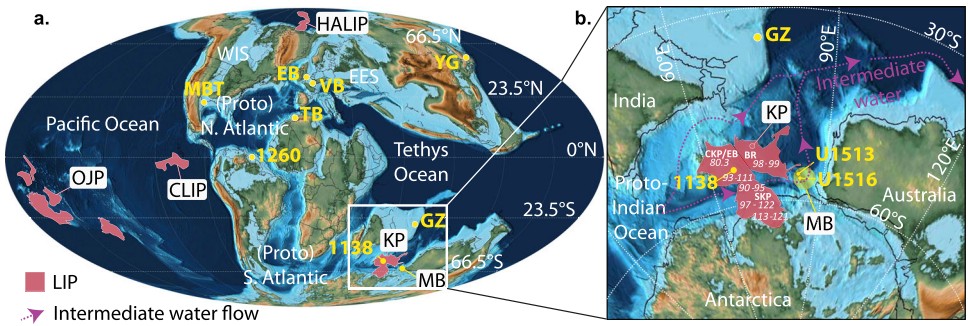

**Fig. 1 | Mid-Cretaceous palaeogeography and large igneous provinces.** Paleogeographic reconstructions of the mid-Cretaceous, created on GPlates using time dependent raster files[72]. **a** Mollweide Projection. Pink shaded areas with a white outline represent modelled extent of active Large Igneous Provinces (LIPs)[73]. HALIP = High Arctic Large Igneous Province; OJP = Ontong Java Plateau; KP = Kerguelen Plateau; CLIP = Caribbean Large Igneous Province; MB = Mentelle Basin; WIS = Western Interior Seaway; EES = Eurasian Epicontinental Sea; GZ = Gongzha Section, Tibet[6]; 1138 = Ocean Drilling Program Site 1138[42]; MBT = Maverick Basin, Texas[22]; VB = Vocontian Basin, France[5]; EB = Eastbourne, UK[23]; YG = Yezo Group, Hokkaido, Japan[5]; 1260 = Site 1260 B, Demerara Rise, Atlantic Ocean[74]; TB = Tarfaya Basin, Morocco[23]. **b** The proto-Indian Ocean region focused on the Kerguelen Plateau and the Mentelle Basin. Dashed purple arrow = mean direction of intermediate water mass (500–1500 m) flow during the Cenomanian[75]. Absolute ages of volcanic activity on the Kerguelen Plateau and named sub-sections of the LIP[13]. NKP = North Kerguelen Plateau; BR = Broken Ridge; SKP = South Kerguelen Plateau; CKP/EB = Central Kerguelen Plateau/Elan Bank.

southern WIS adjacent to the Caribbean, compared to sites in the Western Pacific and the central WIS[6].

Sedimentary mercury (Hg) is potentially a more direct proxy than Os isotopes for LIPs[22–26] as it is produced in large quantities by volcanism (accounting for 20%–40% of modern natural Hg atmospheric emissions[27]), and exhibits a short residence time in the atmosphere (<1 year[28]) and oceans (decades to centuries[29]). During a LIP event, emissions of mercury into the atmosphere would have been orders of magnitude beyond usual background levels[30]. The short timeframe of atmospheric mixing may allow for globally preserved heterogenous signals from massive subaerial volcanic Hg emissions, whilst submarine volcanism sourced from hydrothermal vents or expelled fluids from reactions between mafic rocks and seawater may cause more regional marine Hg signals[22,24], requiring the site of eruption to be proximal to the sedimentary record in order to be detected[23]. After atmospheric emission, precipitation and dust-fall removes Hg, which is either deposited directly into the ocean, or first on land where it is cycled through organic matter and clay and later released into oceans via riverine transport[22]. Following marine hydrothermal emission, Hg is likely to remain within the ocean water column, either staying proximal to the source or transported distally with ocean currents. Modelling suggests the majority of marine Hg is sequestered in sediments within <1 kyr[30,31], usually taking the form of oxidised $Hg^{2+}$ which has a high affinity to sulfur-containing functional groups and acidic functional groups present in dissolved and particulate organic matter (OM)[32]. The relatively modest OAE2 increases in Hg and Hg relative to total organic carbon (Hg/TOC) in the WIS and (proto-)North Atlantic is therefore surprising if OAE2 was triggered by the nearby CLIP[22]. Furthermore, the modest Hg/TOC values from OAE2 sections in the Tethys Ocean provide no evidence for significant volcanic-related excess Hg[23]. Whilst evidence such as lead (Pb) isotopes[33] and pyroclastic and basalt deposits[14] suggest subaerial eruptions from the Caribbean and Madagascar flood basalts and the HALIP around the time of OAE2, the lack of a global Hg signature is contrary, given the likelihood of global atmospheric Hg mixing and deposition in such eruptions. On the other hand, MCE initiation is understudied, and no consensus on MCE related LIP involvement has yet been reached[22,34,35].

Here, we measure Hg and Hg/TOC at two high palaeolatitude (~60°S) sites in the Mentelle Basin[36] – located relatively close to the mid-to-late Cretaceous (~122–90 Ma) Kerguelen LIP (Fig. 1, spanning ~50–62°S) – in order to constrain the possible source and timing of large volcanic episodes associated with OAE2 and the MCE. International Ocean Discovery Program (IODP) Sites U1516 and U1513 contain expanded Cenomanian sections in a relatively deep water bathyal setting[36], ideal for Hg analysis. To compare volcanic pulses with regional and global environmental changes, we stratigraphically constrain OAE2 and the MCE within an existing biostratigraphic framework using $\delta^{13}C$ measurements of bulk rock carbonate, TOC, and single species of benthic foraminifera, and then constrain weathering products using bulk sedimentary neodymium and strontium isotopes ($\varepsilon$Nd and $^{87}Sr/^{86}Sr$)[37]. We also constrain environmental changes with $\delta^{13}C$ and $\delta^{18}O$ data, benthic foraminiferal assemblages, and other published productivity proxies. We find higher Hg and Hg/TOC values associated with OAE2 initiation than in Caribbean records, indicating volcanic activity near the Mentelle Basin, such as the Kerguelen LIP. This is supported by an increase in the deposition of radiogenic $\varepsilon$Nd, consistent with the weathering and erosion of a recently uplifted Kerguelen Plateau. Based on Hg/TOC pulse timings, we suggest volcanic emissions played a significant role in triggering changes in the climate and carbon cycle, as well as regulating the phases of OAE2. In contrast, our Hg data demonstrate that there is little evidence that the MCE began with a volcanic event near the Mentelle Basin. We infer high productivity and possible upwelling throughout OAE2 in the Mentelle Basin, based on benthic foraminiferal assemblages and bulk sediment isotopes, similar to Northern Hemisphere records[22,34,35].

## Results and discussion
### Mentelle basin OAEs
We correlate Sites U1513 and U1516 (Fig. 2; Supplementary Fig. 1) with new tie points (A–I) using a combination of $\delta^{13}C_{(carbonate)}$, $\delta^{13}C_{(organic)}$, Natural Gamma Radiation (NGR), TOC, XRF-Ca counts, and published nannofossil and foraminifera datums ranging from Albian to Turonian (Methods). The sites are ~40 km apart and record similar environmental and palaeoceanographic histories spanning the MCE and OAE2. Both are characterised by a broad transition from alternations of (nannofossil-rich) claystones and chalks with relatively higher TOC in the Albian and Cenomanian, punctuated by rare and thin organic-rich shales, to chalk and nannofossil-rich clay in the Turonian with lower TOC. Both the MCE and OAE2 are clearly defined by positive excursions in $\delta^{13}C$ and NGR (Fig. 3).

As $\delta^{13}C_{(carbonate)}$ and $\delta^{13}C_{(organic)}$ records do not exhibit the clear characteristic long-term positive CIE often defining OAE2, due in part to a low carbonate horizon and likely changing sources of bulk material, we constructed a composite benthic foraminiferal isotope record across OAE2 for Site U1516 (Fig. 3) using individual species and correcting for species-specific offsets (Methods, Supplementary Data 1,

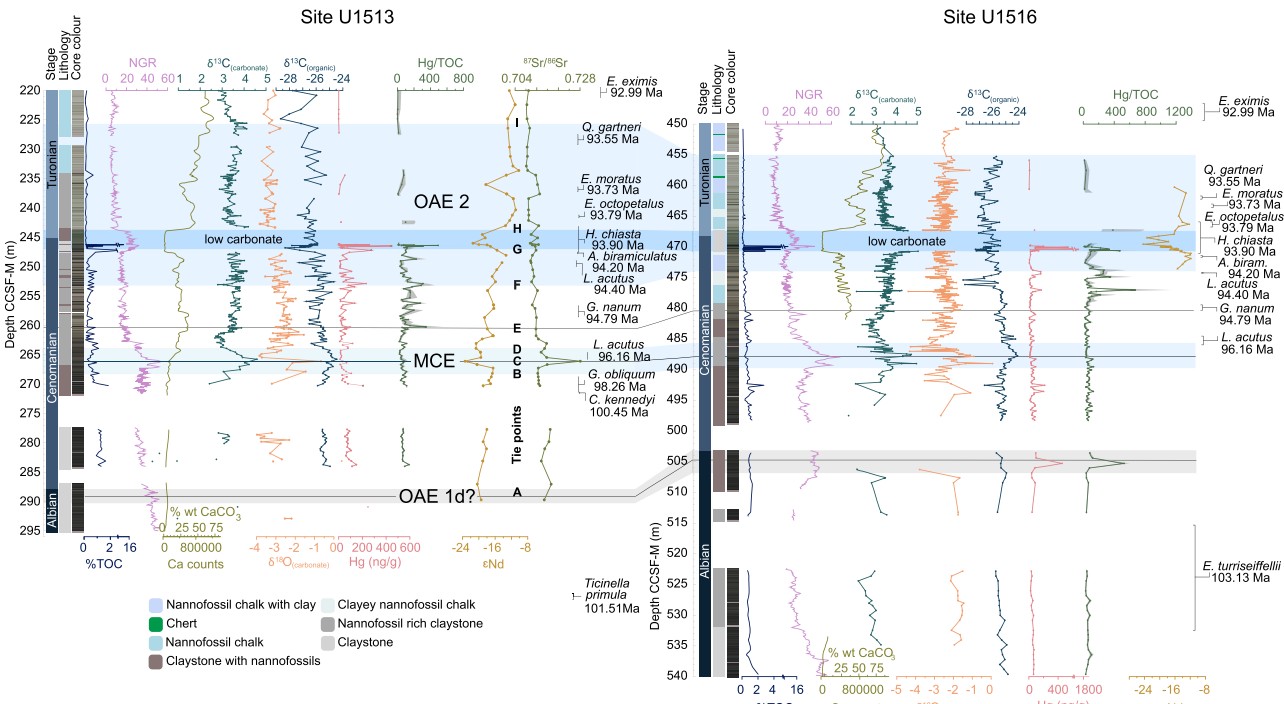

**Fig. 2 | Sites U1513 and U1516 correlation.** Planktonic foraminifera and calcareous nannofossil bio-events[38,39] constrain the timing of Oceanic Anoxic Event 2 (OAE2) and the Mid-Cenomanian Event (MCE). Core lithology, colour and Natural Gamma Radiation (NGR) from International Ocean Discovery Program Expedition 369 Proceedings[36]. Calcium (Ca) counts (X-Ray Fluorescence (XRF) data)[69] correlate with measured % calcium carbonate (CaCO₃), and the relationship between them is used to create a scale for %CaCO₃. CCSF (m) = core composite depth below sea floor

in meters. % total organic carbon (TOC), δ13C(organic), mercury (Hg) and Hg/TOC, 87Sr/86Sr and εNd presented in this study. δ13C(carbonate) is a compilation of data from this and other recent studies[38,39]. Light blue shading represents the MCE and OAE2, with darker blue shading indicating the low carbonate zone in OAE2. A light grey shaded area indicates the possible location of OAE 1d (see Supplementary Materials for identification of OAEs). Grey shading in Hg/TOC data represents uncertainty – see details in methods for calculations.

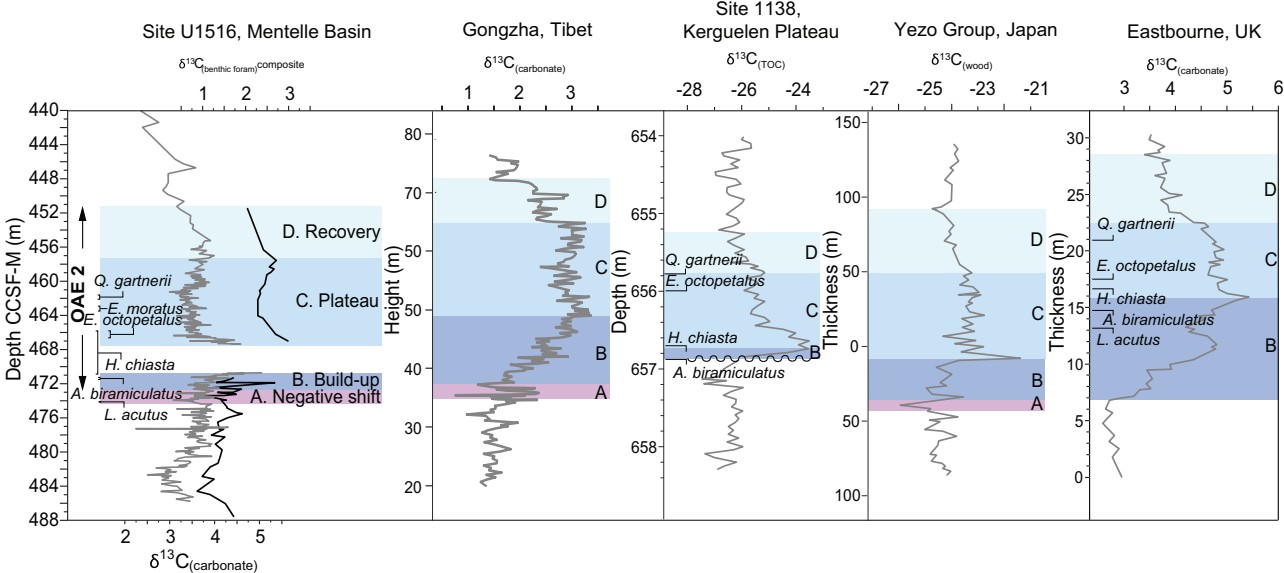

**Fig. 3 | Refined Oceanic Anoxic Event 2 (OAE2) carbon isotope stages in the Mentelle Basin.** Refined carbon isotope stages from site U1516 in the Mentelle Basin, Australia, supported by biostratigraphy[38,39]. Bulk carbonate (grey) and benthic foraminifera δ13C data (black) are shown for the Mentelle Basin – the latter showing a prolonged positive excursion compared to bulk isotope data. Colour

bands indicate different carbon isotope stages of OAE2 that are used to establish correlations of the OAE2 records. Data from the Mentelle Basin (this study), Gongzha, Tingri, southern Tibet[6], Ocean Drilling Program (ODP) Site 1138 (Kerguelen Plateau)[42], Yezo Group, Japan[5] and Eastbourne, UK[76].

Supplementary Figs. 2 and 3). δ13C(benthic) and δ18O(benthic) composite curves reflect bottom water values, and broadly support δ13C(carbonate) as most likely dominated by a surface water signal from mixed nannofossil and planktonic foraminifera. A negative δ13C(carbonate) 'pulse'

of ~0.75‰ at U1513 and U1516 (tie point F in Fig. 2, phase A in Fig. 3) is also present in OAE2 records[5,6]. This is followed by the OAE2 'build-up' phase B, a positive CIE which is only partly present due to very low carbonate content (from tie point G). The peak CIE is not recorded at

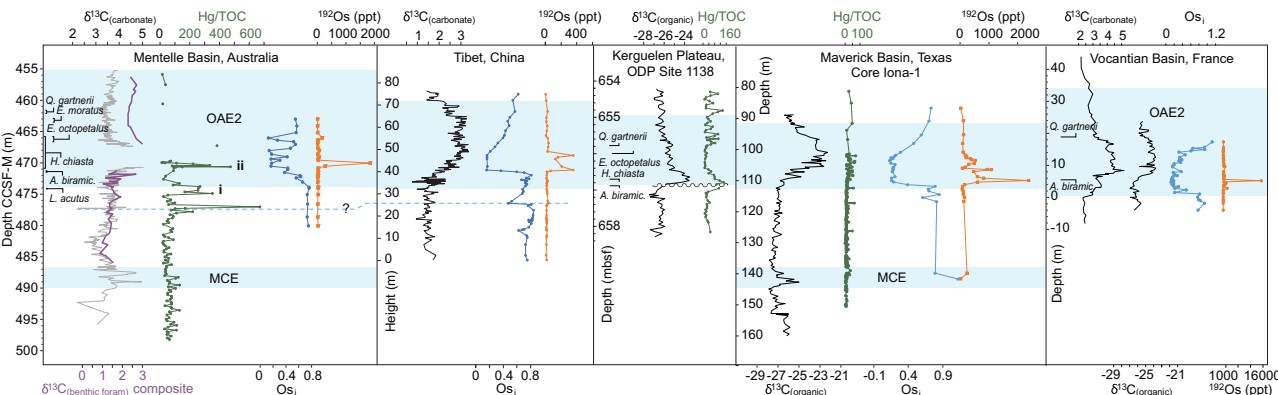

**Fig. 4 | Volcanic signals across Oceanic Anoxic Event 2 (OAE2) and the Mid-Cenomanian Event (MCE).** Hg relative to total organic carbon (Hg/TOC), $\delta^{13}C$ and osmium (Os) isotope records of OAE2 and the MCE from the Mentelle Basin (proto-Indian Ocean)[52], Maverick Basin (Western Interior Seaway)[18,22], Tibet[6], the Kerguelen Plateau[23,42] and Vocontian Trough, south-east France[16,23]. Hg/TOC values in the Mentelle Basin are an order of magnitude higher than at Northern Hemisphere sites

across OAE2, indicating a proximal source of volcanism to southwest Australia. Data from the Kerguelen Plateau are incomplete due to a hiatus during OAE2 onset – precluding direct comparison with the Mentelle Basin, and making a Kerguelen Plateau source of Hg possible, despite generally low Hg/TOC values in the upper portion of the OAE2 record from Ocean Drilling Program (ODP) Site 1138 (Kerguelen Plateau).

---

either site due to low carbonate concentrations, but above this interval (from tie point H) elevated $\delta^{13}C_{(benthic)}$ clearly indicates the OAE2 'plateau' phase C, in contrast to the rapid decline in $\delta^{13}C_{(carbonate)}$[43]. This is supported by biostratigraphy[38,39], as *E. octopetalus* first occurrence (FO) and *Q. gartnerii* FO are in OAE2 phase C at sites on Kerguelen Plateau and Eastbourne[23], UK. The offset of $\delta^{13}C_{(carbonate)}$ from $\delta^{13}C_{(benthic)}$ may be due to a change in the composition of the bulk carbonate (e.g. nannofossil assemblage shift) or change in surface water mass. A gradual decline of $\delta^{13}C$ in both bulk and benthic foraminiferal records shows the start of the 'recovery' phase D.

### Sedimentary Hg, and Nd and Sr isotopes reflect Kerguelen LIP activity at OAE2

Both absolute Hg concentrations and Hg/TOC in the Mentelle Basin (Fig. 2) show significant pulsed enrichments, broadly mirroring one another in association with the lead-up to and early OAE2. At site U1513, pulsed spikes of Hg are as high as 275 and 445 ppb and Hg/TOC reach values of 311 and 491 ppb/wt%. At Site U1516 Hg pulses are 536 and 1609 ppb, with Hg/TOC values of 474 and 383 ppb/wt%. These are significantly higher than baseline values throughout the Albian and lower Cenomanian (with values of Hg ~50 ppb and Hg/TOC < 100 ppb/wt%), and also much higher than Hg concentrations or Hg/TOC values from any other OAE2-bearing strata. An episode of elevated but minor single-point spikes occurs in the lead-up to OAE2 (above tie point E), a significant long-lived Hg/TOC spike is present during the negative $\delta^{13}C$ 'pulse' (tie point F in Fig. 2, 'i' in Fig. 4), and a second major spike occurs just after the onset of the low carbonate horizon (tie point G in Fig. 2, 'ii' in Fig. 4). We conclude these Hg pulses originated predominantly from volcanic activity rather than transported terrigenous material common in marginal settings[31], or from buried redox fronts[23]; as our sites were in deep water distal to the shoreline[36], there is no correlation between Hg and proxies for terrigenous input[40] ($\varepsilon$Nd and K/Al; Supplementary Fig. 4), and no turbiditic sedimentary features are reported[36]. Furthermore, although recent work has suggested euxinic conditions can result in the overprinting of sedimentary Hg[23], we report a lack of isorenieratene indicating no photic zone euxinia (Supplementary Fig. 5), and benthic foraminiferal assemblages indicative of oxic environments below the OAE2 low carbonate interval (Fig. 5). Hg measurements across possible anoxic episodes such as TOC spikes (lower in the low carbonate interval) may therefore be taken as minimum values.

Whilst LIP activity is widely agreed as a probable trigger for OAE2, the location of the LIP most likely to have influenced this anoxic event

is still debated. However, Southern WIS and Demerara Rise records over OAE2 show relatively low Hg (mostly < 100 ppb) and Hg/TOC elevations (mostly < 50 ppb/wt%) (Fig. 4)[22,23] compared to our data from the Mentelle Basin, suggesting a Southern Hemisphere volcanic source for the Mentelle Basin mercury spikes. The highest Hg/TOC values in the Mentelle Basin are more similar to records spanning the end-Triassic Central Atlantic Magmatic Province and OAE1d (another Cretaceous OAE, occurring ~100.2 Ma)[26,41] (> 200 ppb/wt%) which, when coupled with their stratigraphic appearance at the beginning of OAE2, point to a nearby Southern Hemisphere volcanic source capable of triggering global climate change (Fig. 1).

The nearby Kerguelen LIP was volcanically active for > 32 myr and although precise dating is relatively limited, recent $^{35}$Ar/$^{34}$Ar dating shows an active eruptive phase of the Central Kerguelen Plateau at $92.8 \pm 1.5$ Ma[13] – overlapping (within error) with OAE2, ~94 Ma. The Hg/TOC record from nearby Kerguelen Plateau Ocean Drilling Program Site 1138 shows higher values than North Atlantic sites, but is missing the critical OAE2 onset interval due to a hiatus[23,42] where we might expect elevated Hg (Fig. 4). Hiatuses are perhaps unsurprising as uplift likely occurred during eruptive phases of the Kerguelen Plateau[43].

To test for changes in sediment provenance consistent with an uplifting landmass from nearby LIP activity, we build on a recent study of Site U1516[37] and measure fine fraction sedimentary $\varepsilon$Nd and $^{87}$Sr/$^{86}$Sr at U1513 (Fig. 2). The <2 $\mu$m fraction values are considered consistent between craton sources and ocean basins[44], and we find similar values in coarser fractions supporting this provenance proxy (Supplementary Fig. 6). A long-term shift to higher (more radiogenic) $\varepsilon$Nd and lower $^{87}$Sr/$^{86}$Sr values occurs in the build-up to (between tie points C and G) and over OAE2, indicative of an increase in the proportion of sedimentary material deposited in the Mentelle Basin from erosion of younger rocks such as the Bunbury Basalts (~130–137 Ma)[45], Naturaliste Plateau (Early Cretaceous)[46] or Kerguelen Plateau (Fig. 6, Supplementary Fig. 7, Supplementary Data 3). In contrast, less radiogenic values occur during the MCE and the low carbonate interval of OAE2 (also measured in U1516)[37] which have been interpreted as resulting from an increased hydrological cycle reactivating continental river systems and eroding the Yilgarn Craton[37]. Given the dramatic sedimentation rate increase of U1513 after the onset of elevated Hg/TOC at ~260 mbsf (averages of 0.3 cm kyr$^{-1}$ below, to 2.2 cm kyr$^{-1}$ above, Supplementary Fig. 8), this extra detrital input must have come from a newly available radiogenic continental source, rather than simply the result of a reduced hydrological cycle and a (necessarily smaller) input of more local Bunbury Basalts. Whether this was from a newly uplifted

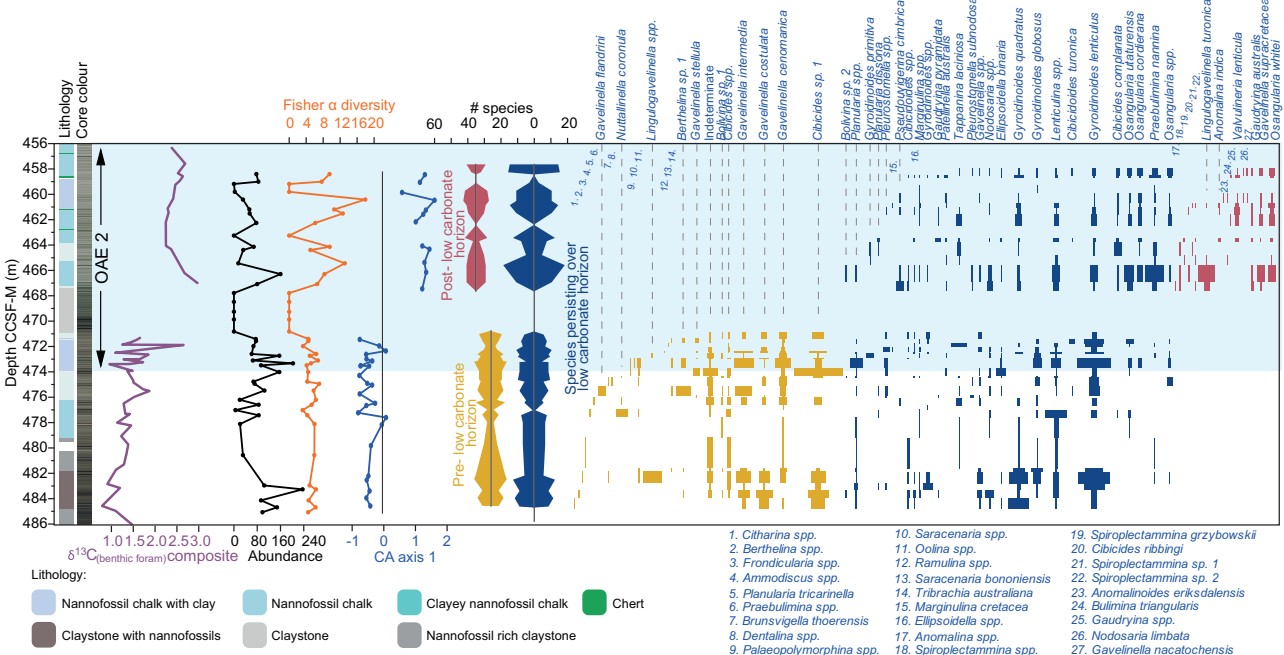

**Fig. 5 | Site U1516 benthic foraminifera assemblage shift over Oceanic Anoxic Event 2 (OAE2).** Range chart, abundance, Fisher α diversity, Correspondence Analysis (CA) Axis 1, and carbon isotopes from benthic foraminifera. A significant shift in assemblage is seen either side of the low carbonate horizon. Axis 1 of the CA correlates with the proportion of infaunal / high productivity species (Supplementary data 2). Thus, high values above the low carbonate horizon suggest high organic carbon flux in the later stages of OAE2.

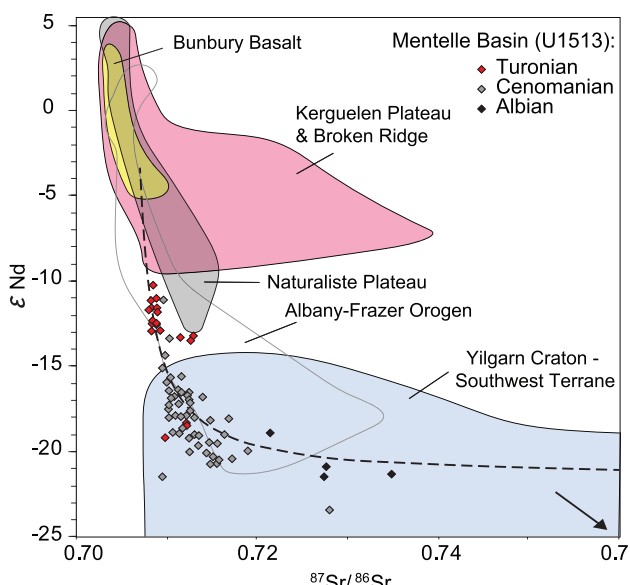

**Fig. 6 | Change in sediment provenance from Albian - Turonian in the Mentelle Basin.** Neodymium (Nd) and strontium (Sr) isotope data from detrital sediments of Site U1513 (this study), Mentelle Basin, and proximal geological terranes. Fields represent the approximate ranges of terrane values based on published literature (Supplementary Data 3). The dashed line represents a mixing calculation between the average values for the Kerguelen Plateau and Yilgarn Craton (Southwest Terrane). Site U1513 Turonian values (red) are approximately concurrent with Oceanic Anoxic Event 2.

Kerguelen Plateau and/or Naturaliste Plateau – the former we suggest as more likely – our data support concurrent Kerguelen LIP activity (rather than the far-field Madagascar LIP; Fig. 1) as the source of the coeval Hg pulses and local increase in the deposition of radiogenic sediments.

Estimating the submarine versus subaerial proportion of Kerguelen Plateau flood basalts is difficult, although extensive evidence exists for both[43]. Volcanic activity associated with the Hg and Hg/TOC spikes across OAE2 in the Mentelle Basin may have been largely marine-based, and therefore more strongly expressed locally, although the modest increases in Hg/TOC in Atlantic records[22] may indicate a partial atmospheric signature. The tholeiitic basalts are thought to have contained significant quantities of sulfur[43], and by implication Hg[27] and, similar to the CLIP, have Pb isotope values[13,43] similar to those measured across OAE2 in central Italy[33]. Kerguelen eruptions would have allowed injection of gasses and ash directly into the stratosphere, which is closer to the Earth's surface at higher latitudes[47]. This likely promoted the global influence of atmospheric volatiles, as such particles and gases have a longer residence time and thus greater dispersal globally in the stratosphere than if injected into the troposphere[43]. Palaeo-currents likely influenced the spread of Hg within the ocean basin, and thus the concentration of volcanically-derived Hg in sediments proximal to the source. The palaeogeography of the Cenomanian/Turonian prevented deep water flow between the proto-Indian Ocean and the proto-South Atlantic, the Pacific and the northern Tethys, and restricted intermediate depth currents connecting these regions (Fig. 1)[48,49]. This makes the marine transport of Hg from any other LIP, such as the Caribbean or Madagascan LIP, to the Mentelle Basin unlikely. Conversely, based on modelled reconstructions of Cretaceous ocean currents, intermediate water may have had a net eastward flow from the Kerguelen Plateau, providing a plausible pathway for marine Hg to enter the Mentelle Basin[49]. Whilst marine transportation of Hg from other active LIPs is unlikely, we acknowledge the possibility that atmospheric emissions from a combination of other sources may have contributed to the initiation of OAE2, in addition to the significant activity of the Kerguelen Plateau. However, the lack of a consistent global Hg signal reduces the significance of atmospheric contributions, relative to the strength of signal in the Mentelle Basin, lending strength to the suggestion of a Southern Hemisphere source of Hg, i.e. the Kerguelen Plateau.

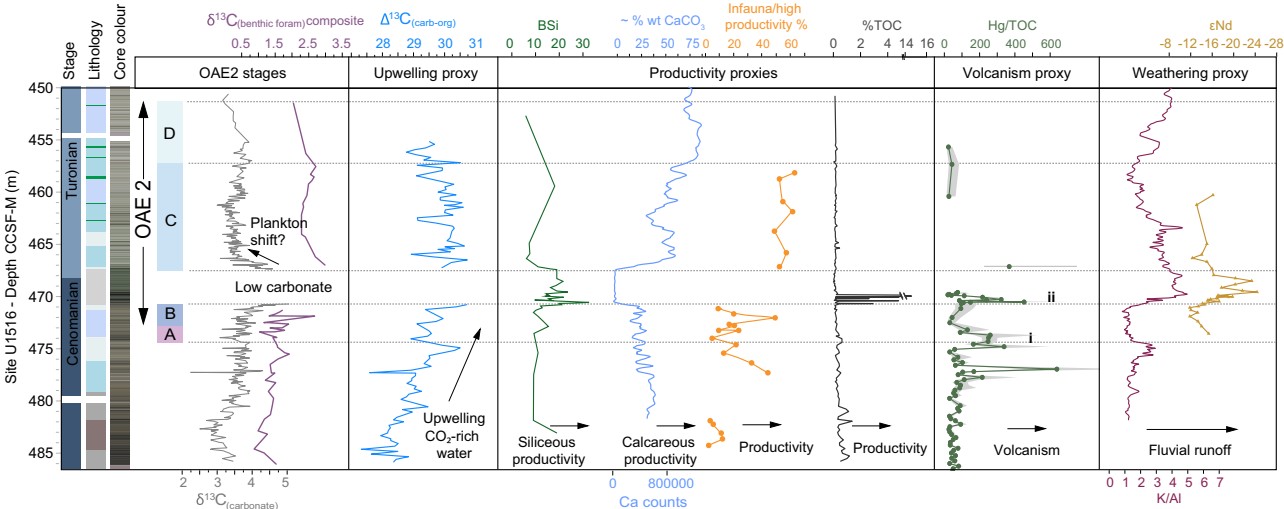

**Fig. 7 | Environmental interpretations across Ocean Anoxic Event 2 (OAE2).**
Summary figure showing environmental interpretations for the Mentelle Basin, Site U1516, during OAE2. Δ13C interpreted as local increased $CO_2$ in the surface waters – either as a result of higher atmospheric $CO_2$ levels or increased upwelling. $\delta^{13}C_{(carbonate)}$ and $\delta^{13}C_{(benthic\ foram)}$ composite show similar trends prior to OAE2, and during phases A and B. After the low carbonate horizon, there is an offset between the two, possibly caused by a change in the dominance or abundance of different planktic groups. Hg relative to total organic carbon (Hg/TOC) data show sporadic enhanced volcanism prior to OAE2, with the first large Hg/TOC signal occurring at OAE2 onset, corresponding with a negative shift in $\delta^{13}C$ for both bulk and composite benthic foraminiferal records. Another large spike in Hg/TOC corresponds with the start of enhanced fluvial runoff, demonstrated by K/Al and εNd data[37]. Productivity is enhanced over OAE2, and remains high during phase C, despite the apparent recovery in $\delta^{13}C_{(carbonate)}$. A particularly intense period of productivity over the low carbonate horizon is suggested by high levels of biogenic silica (BSi)[52].

## Evolution of Cenomanian volcanism from Hg and Os records

There are slightly elevated levels of Hg and Hg/TOC in the lower part of the MCE, relative to background values, at both Mentelle Basin sites (Fig. 2). Although these changes are small (~50–100 ppb/wt%), they are similar to those found in other sites that capture the MCE in the proto-North Atlantic and WIS[22]. In the Mentelle Basin Hg and Hg/TOC values are slightly elevated at the onset of the MCE CIE, before dropping to background levels mid-way through. However, this is in contrast to records from the Maverick Basin (southern WIS) and Demerara Rise (proto-North Atlantic), where elevated Hg/TOC values occur mid-way through the CIE[22]. Despite an absence of low $Os_i$ values, indicating volcanism may not have been the primary driver[8,12,13,20], our Hg records suggest volcanism may have had a role in triggering the MCE, although the source was likely distal to the Mentelle Basin.

Notable spikes in both Hg and Hg/TOC occur at Sites U1513 and U1516. These enrichments exceed Hg concentrations of 400 ppb, and 450 ppb/wt% for Hg/TOC, and are particularly pronounced in the lead-up to and early stages of OAE2, in contrast to the comparatively lower baseline values during the Albian and lower Cenomanian periods where Hg levels are ~50 ppb and Hg/TOC < 100 ppb/wt%. Os records over OAE2 from the Maverick Basin (WIS)[22], Tibet[6], and the Vocontian Trough, south-east France[50], are somewhat different from Hg/TOC records (see Fig. 4 for comparison of Hg and Os data at these sites), in part due to their geographic separation and sedimentary weathering inputs, and in part due to the much longer ocean residence time for Os (~10 kyr)[51]. It is therefore possible that submarine LIP activity may demonstrate a global $Os_{(i)}$ excursion, whilst Hg is more restricted in its submarine dispersal, showing only local/regional excursions. Thus, any submarine LIP activity, not just the Kerguelen Plateau, may be influencing $Os_{(i)}$ data globally. Conversely, the Hg record would be restricted to nearby strata, necessitating a proximal source. Furthermore, it is also challenging to correlate precisely between sites because biostratigraphy has some uncertainty on short timescales, and the characteristic CIE used to define the OAE2 onset is usually measured on bulk sediment. Indeed, $\delta^{13}C_{(organic)}$ and $\delta^{13}C_{(carbonate)}$ records commonly diverge from one another. $\delta^{13}C_{(organic)}$ records frequently exhibit a delayed or absent positive CIE (e.g., Mentelle Basin, Tibet,

Vocantian Basin; Fig. 4), whilst important sites from the WIS are based solely on $\delta^{13}C_{(organic)}$. A recent study on the WIS Iona-1 site used $\delta^{13}C_{(organic)}$ data to correlate a positive/negative $Os_i$ shift to early OAE2 and used this as evidence for CLIP activity[6], but we note alternative correlations exist[18]. Furthermore, it is likely that successive Kerguelen LIP eruptive phases released varying relative proportions of Os and Hg, as considerable geochemical variability is detected between different eruptive phases due to changes in relative melt incorporation of plume, continental and oceanic crust[43]. The numerous Hg/TOC pulses between tie points E and F (Fig. 2) document early volcanic activity, and although this is not recorded in Os isotopes from U1516[52] (possibly due to low resolution), precursor negative $Os_i$ excursions do occur in Tibet, the WIS, and Japan[6,16,50] (Fig. 4). This precursor OAE2 volcanism was likely relatively minor as it occurred before major climate changes associated with OAE2, whilst Hg 'spike i' was relatively major as it is more stratigraphically extensive (multiple data points at both sites, Fig. 2), occurs at the beginning of the $Os_i$ excursion in U1516 (Fig. 4), and is associated with the initial negative $\delta^{13}C$ 'pulse' of OAE2 phase A (Figs. 2–4). A similar sharp negative shift has also been documented in England[53], the southeast North Atlantic[54], the western Pacific[55] and the WIS[56], and has been suggested as linked to LIP-related carbon release at the onset of OAE2[5].

## Volcanism-induced palaeoenvironmental change over OAE2

Kerguelen LIP carbon release[43] likely caused ocean warming, which in the Cretaceous may have disrupted the thermocline, triggering upwelling of nutrient-rich waters to sustain enhanced productivity[57]. Between 478 and 474 m in U1516 (Fig. 7), the dominance of opportunistic planktonic foraminifera *Microhedbergella* has been interpreted as signifying enhanced nutrient runoff and likely upwelling[38], and our benthic foraminiferal assemblage data supports changing organic carbon flux with varying infaunal/high productivity species ranging from 5 to 40%. To test for upwelling, we calculated the difference between measured $\delta^{13}C_{(organic)}$ and $\delta^{13}C_{(carbonate)}$ values in the same sample ($\Delta^{13}C$) as a proxy for relative changes in surface ocean $CO_2$[58]. Our data shows a gradual and sustained divergence (increased $\Delta^{13}C$) in the lead up to and early OAE2 phases A and B (Fig. 7), supporting

enhanced upwelling bringing $CO_2$-rich water to the surface, and/or increasing atmospheric $CO_2$[59], possibly linked to volcanism.

The second significant Hg and Hg/TOC 'spike ii' (Fig. 4) occurs near the base of the low carbonate interval, and likely identifies a significant episode of LIP volcanism due to its association with severe environmental change in the Mentelle Basin (e.g., ocean acidification[52] and an enhanced hydrological cycle[37]), and a pulse in Os with low $Os_i$ values in U1516 likely global in nature (Fig. 4). Enhanced productivity, as evidenced by TOC spikes, biogenic silica[52], and radiolarians[38] (Fig. 7) occurred with enhanced upwelling which would have further increased $\delta^{13}C_{(organic)}$ (Fig. 2). Above the low carbonate interval (OAE2 phase C), benthic foraminiferal assemblages contain a greater proportion of infaunal/high productivity species, increasing % $CaCO_3$, and high productivity *Microhedbergella* and "*Globigerinellioides*"[38], despite the presence of organic poor sediments. Due to low TOC, Hg data are absent, except one point in U1516 which suggests at least one possible further episode of volcanism through phase C (Fig. 7); this is supported by Os and $Os_i$ pulses that occur above the low carbonate interval, suggested as from Kerguelen volcanism[52].

Our work highlights the utility of using multiple volcanic proxies in diagnosing causal mechanisms for past global warming events, constraining the provenance of marine sediment, and the critical importance of measuring species-specific isotopes in refining global OAE correlations. Specifically, we refine a geochemical signature for Kerguelen LIP volcanism, and identify it as a likely trigger for OAE2. Future study of Kerguelen Plateau eruptive phases, Hg isotopes and earth system modelling will provide insights into LIP emissions and climate interactions, and identification of past tipping points within the climate system.

## Methods

### Organic carbon isotope analysis

Sediment samples for isotope analysis on the organic fraction were processed by disaggregating around 1 g of sample in a 500 ml glass beaker containing 5% HCl solution for 24 hours to remove carbonate material. This solution was then diluted to 500 ml and rinsed three times with deionised water to remove any remaining acid. After carefully tipping off the majority of deionised water from the final rinse (< 50 ml), the samples were dried at 40 °C and subsequently powdered and homogenised in an agate pestle and mortar.

Carbon isotopic ($\delta^{13}C$) analysis of samples was carried out at the National Environmental Isotope Facility at the British Geological Survey (UK) using an Elementar vario ISOTOPE cube elemental analyser (EA) coupled to an isoprime precisION isotope ratio mass spectrometer (IRMS) with an onboard centrION continuous flow interface system. The EA inlet converts organic materials in solid sample matrices into pure gases via high-temperature combustion. The post-combustion gas mixture is then separated and focused into individual molecular species for quantitative analysis of nitrogen and carbon content and are then passed online to the IRMS for the determination of their stable isotope composition. Carbon isotope data are reported in delta ($\delta$) notation in per mille (‰) relative to the international reference scale VPDB. Carbon isotope ratios were normalised to the VPDB scale using a multi-point calibration incorporating organic analytical standards B2162 (spirulina algae, Elemental Microanalysis Ltd.; −18.7‰, in-house value), B2151 (soil, Elemental Microanalysis Ltd.; −28.9‰), and B2213 (spruce powder, Elemental Microanalysis Ltd.; −25.4‰), and a laboratory working standard (BROC3, −27.6‰). The reference materials BROC3 and B2162 have been calibrated for $\delta^{13}C$ using IAEA-CH-6 (−10.5‰), USGS54 (−24.4‰), USGS40 (−26.4‰), and B2174 (urea, Elemental Microanalysis Ltd.; −36.5‰). External precision (1σ) for the within-run standards and sample repeats was <0.1‰. BROC3 (41.3 %C and 4.9 %N) was used to calculate the carbon and nitrogen elemental content of samples.

### Carbonate isotope analysis

Sediment samples for isotope analysis on carbonate (calcite) were processed by disaggregating around 1 g of sample in a 500 ml glass beaker containing 5% sodium hypochlorite (NaClO) solution for 24 hours to oxidise any reactive organic matter (OM). This solution was then diluted to 500 ml and rinsed three times with deionised water to remove any remaining NaClO or oxidised OM. After carefully tipping off the majority of deionised water from the final rinse (< 50 ml), the samples were dried at 40 °C and subsequently powdered and homogenised in an agate pestle and mortar.

Each carbonate sample was weighed into a glass vial to provide ~10 mg $CaCO_3$ and placed into a glass reaction vessel containing anhydrous phosphoric acid ($H_3PO_4$), which is attached to a glass vacuum line and evacuated at the National Environmental Isotope Facility at the British Geological Survey (UK). Once a sufficient vacuum pressure had been achieved ($< 8 \times 10^{-5}$ mbar), the vessels are sealed and transferred to a water bath at 25 °C to equilibrate for at least 15 min. The vessels are then overturned, and the sample reacted with the phosphoric acid.

The vessels are then returned to the water bath and left to react for at least 16 h at a constant 25 °C. After allowing enough time for a complete reaction, any remaining water vapour is removed from the liberated $CO_2$ by passing the gas through a cold trap held at −90 °C on the vacuum extraction line. The purified $CO_2$ is subsequently transferred and frozen into collection vessels submerged in liquid nitrogen and evacuated to $<2 \times 10^{-5}$ mbar to remove any other gaseous fraction.

The evolved $CO_2$ was analysed using a VG Optima or Thermo MAT 253 dual inlet mass spectrometer relative to a reference $CO_2$, where stable isotope measurements are made on $CO_2$ from both the sample and within-run carbonate laboratory standards (MCS and CCS). $\delta^{13}C$ and $\delta^{18}O$ are calculated from the mass ratios 45/44 and 46/44, respectively, relative to the Vienna Pee Dee Belemnite (VPDB) scale using a single-point anchoring procedure based on calibrated $\delta^{13}C_{MCS-VPDB}$ and $\delta^{18}O_{MCS-VPDB}$ values (via NBS and IAEA international reference materials). A correction is applied to the 45/44 and 46/44 ratios for the minor contribution from $^{17}O$ on the 45 ($^{12}C^{17}O^{16}O$) and 46 ($^{13}C^{17}O^{16}O$) ion beams[60]. A fractionation factor is also applied to $\delta^{18}O$ as although all carbon is transferred to the evolved $CO_2$ during reaction with phosphoric acid, only two-thirds of the oxygen is collected. The acid fractionation factor is constant in this case as the oxygen isotope fractionation between the evolved $CO_2$ and original mineral is temperature-dependent and the reaction is controlled at 25 °C using the water bath. The fractionation factor ($\alpha$) between $CO_2$ and calcite during reaction with phosphoric acid at 25 °C is 1.01025[61]. Using $\alpha_{CO_2-calcite}$ the $\delta^{18}O$ of the original calcite is then calculated using $1.01025 = [1000 + \delta_{CO_2}] / [1000 + \delta_{calcite}]$[62]. The analytical reproducibility calculated from the standard deviation (1σ) of the within-run laboratory standards is typically < 0.1‰ for both $\delta^{18}O$ and $\delta^{13}C$.

### Hg analysis

Hg analysis was carried out on 259 bulk sediment samples (145 from Site U1516 and 114 from Site U1513) by an RA 915 F Lumex Portable Mercury Analyser at the Camborne School of Mines, University of Exeter. Methods were adapted from previous studies[24]. After decarbonisation, to allow consistent normalisation with TOC, approximately 50 mg of sediment powder were measured into a quartz measuring boat and its precise mass determined. Samples were heated in the Pyrolyzer to ~700 °C to volatilise the Hg within the sample. Following this, gaseous Hg was transported into the Analyzer and abundance was measured, providing the abundance mass of Hg as parts per billion.

The Analyser was initially calibrated by measuring 20, 30 and 40 mg of the following international standards: NIST 2682b (37.3 ng/g), NIST 2693 (108.8 ng/g), NIST 2709 (1400 ng/g) and NIST

2711 (6250 ng/g). The resulting calibration curve yields a correlation factor of 0.9996. An additional calibration curve for lower Hg values was then created by repeating NIST 2682b and NIST 2693 five times, which resulted in a correlation factor of 0.9936. An internal standard called PM43, calibrated to these international standards, was used the beginning and end of every run and after every 8 samples during the analytical run to ensure continuity. PM43 has an average Hg concentration of $107 \pm 3$ ppb and is composed of powdered and homogenised mudrock from a horizon of the Whitby Mudstone Formation, Port Mulgrave, Yorkshire. For this study, the mean value for PM43, recorded from the beginning of each analytical run and the 8-sample interval within-run repeats, was $108.1 \pm 3.0$ ppb (72 measurements). The average standard deviation of repeated samples was 5.6 ppb (39 repeats). Following Kender et al.[24] the minimum possible value for Hg/TOC is represented by: (sample Hg − 1 standard deviation) / (TOC + 1 standard deviation). The highest possible Hg/TOC value is calculated as: (sample Hg + 1 standard deviation) / (TOC − 1 standard deviation). The standard deviation of analytical uncertainty for Hg values is 3 ppb, and for TOC is 0.1%.

For both records, sedimentary Hg shows highly fluctuating values, some samples with Hg concentrations below the detectable limit, to 1614 ppb. An organic carbon association for this sedimentary Hg is supported by the relationships between Hg and TOC across all samples from Sites U1513 and U1516, and the very low Hg values in samples with low TOC (Supplementary Fig. 9A–C). Poor relationships were found ($R^2 < 0.08$) between Hg concentration and XRF counts for Al, S and Fe (Supplementary Fig. 9 D–J).

## Sr and Nd isotope analysis
Samples were weighed into Savillex teflon beakers and leached in warm 10% acetic to remove carbonate. After centrifuging and discarding the leachate, the samples were washed and centrifuged twice in mQ water, dried and reweighed, before spiking with $^{84}Sr$ and mixed $^{149}Sm$-$^{150}Nd$ isotope tracers. The samples were dissolved in 16 M $HNO_3$ / 29 M HF then converted to chloride form, redissolved in calibrated 2.5 M HCl in preparation for column chemistry, and centrifuged.

Sr and a bulk rare earth element (REE) fraction were separated using cation exchange columns containing Dowex AG50x8 resin. Nd fractions were separated using EICHROM LN-SPEC ion exchange columns.

Nd fractions were run in three sessions, using both Thermo Scientific Neptune MC-ICPMS and Triton TIMS instruments. Data are normalised to $^{146}Nd/^{144}Nd = 0.7219$. In the first two sessions on the Neptune instrument, operated in static multicollection mode, multiple replicate analyses of the JNd-i standard gave mean values of $0.512100 \pm 0.000029$ (1-sigma, $n = 17$) and $0.512084 \pm 0.000016$ (1-sigma, $n = 10$). In the third session on the Triton mass spectrometer, operated in multidynamic mode, JNd-i gave mean results of $0.512094 \pm 0.000005$ (1-sigma, $n = 3$). Results are quoted relative to a value of 0.512115 for this standard. Seven analyses of the BCR-2 rock standard run with the samples gave a value of $0.512618 \pm 0.000012$ (1-sigma).

Sr fractions were loaded onto outgassed single Re filaments using a TaO activator solution and analysed in a Thermo-Electron Triton mass spectrometer in multi-dynamic mode. Data are normalised to $^{86}Sr/^{88}Sr = 0.1194$. Samples were run in three sessions. The NBS987 standard in each session gave mean values of $0.710262 \pm 0.000008$ (1-sigma, $n = 13$), $0.710259 \pm 0.000006$ (1-sigma, $n = 7$), and $0.710263 \pm 0.000005$ (1-sigma, $n = 10$). Sample data is normalised using a preferred value of 0.710250 for this standard. Five analyses of the BCR-2 rock standard run with the samples gave a value of $0.705015 \pm 0.00007$ (1-sigma).

## Benthic foraminiferal analyses
A total of 42 samples from Site U1516 were investigated for benthic foraminiferal assemblages (Fig. 5). Two to three samples per core section were collected from Cores U1516D-2R to U1516D-5R, and one to two samples per core section were collected from Cores U1516C-32R to U1516C-35R-1W. All samples of approximately 20–30 cc were disaggregated completely by screen-washing at 63 μm, and oven dried at low temperature following standard procedures. All specimens were picked into cardboard reference slides from the >63 μm fraction and identified to species level where possible. Preservation is moderate to poor and in some cases identification could only be made to the generic level. The suprageneric classification follows Tappan and Loeblich[63] while the taxonomic concept mainly follows Belford[64], Hanzlikova[65], and Holbourn and Kuhnt[66]. Benthic foraminiferal ranges are illustrated in Supplementary data 2, and names and references in the taxonomic list. Key species were imaged after carbon coating using a TESCAN VEGA3 GMU SEM at the Penryn Campus, University of Exeter (Supplementary Fig. 10). The figured specimens are stored at the Camborne School of Mines, Cornwall, UK – contact s.kender@exeter.ac.uk for access.

Statistical Correspondence Analysis (CA) and Fisher's Alpha (α) diversity was carried out on the entire benthic foraminiferal dataset using the software PAST v. 2.17c[67]. Fisher's Alpha is used because it is theoretically independent of sample size, as it estimates the area beneath a smoothed curve of a parametric species abundance distribution[68]. Diversity averaged 14 species per sample, and most species of cosmopolitan distribution ranging through water depths of outer neritic to upper bathyal (Supplementary Table 3). Species were grouped according to their palaeoecology (Supplementary Table 3), with some information from $\delta^{13}C$ (Supplementary Fig. 11), which was used to create the 'Infauna/high productivity %' plot in Fig. 6. CA indicates the most significant assemblage shift occurs either side of the low carbonate horizon, with 26 species restricted to below this, 17 species appearing after, and 28 survivor species persisting throughout the record (Fig. 5). Axis 1 of the CA correlates well with the proportion of infaunal / high productivity species (Supplementary Table 3, Supplementary Fig. 12).

## Sample processing and foraminiferal stable isotopes
Measurements of $\delta^{13}C$ and $\delta^{18}O$ of the material were carried out on all 42 samples across 10 species of benthic foraminifera, as there was no single species that occurred in all samples at the required abundance. Contingent on the species, between 1 and 10 specimens per species were measured for each sample (approximately 60 – 100 micrograms of carbonate). Measurements were carried out at the at the National Environmental Isotope Facility at the British Geological Survey (UK), using an IsoPrime 100 dual inlet mass spectrometer with a MultiCarb preparation device. The samples were loaded into glass vials and sealed with septa, after which the vials were evacuated and reacted with anhydrous phosphoric acid at 90 °C. The evolved $CO_2$ was collected cryogenically for 15 min and passed on-line to the mass spectrometer. Isotope values ($\delta^{13}C$, $\delta^{18}O$) are expressed as per mille deviations of the isotopic ratios ($^{13}C/^{12}C$, $^{18}O/^{16}O$) calculated to the Vienna Pee Dee Belemnite (V-PDB) scale using a within-run laboratory standard (KCM) calibrated against NBS-19. The calcite-acid fractionation factor applied to the gas values is 1.00798. Due to the long run time, a drift correction was applied across the run, calculated using standards that bracketed the samples. The Craig correction[60] was also applied to account for $^{17}O$ contribution. The average analytical reproducibility of the KCM standard is < 0.1‰ for $\delta^{13}C$ and $\delta^{18}O$.

## Composite benthic isotope record
To construct a consistent composite isotope record showing changes in ocean chemistry, we calculated species specific isotopic offsets. Between 471.4 and 479.8 m CCSF-M (a period with little fluctuation in isotopic composition) the average $^{13}C$ and $^{18}O$ values were calculated for *Cibicides*, *Gavelinellinae*, *Gyroidinoides* and *Osangularia utaturensis* (Supplementary Figs. 2 and 3). Using these averages, the offset between *Cibicides* and each of the other groups was calculated, and this offset was then applied to all data for each species. Where isotope

values for multiple species were available for one depth horizon, the average of these was calculated for use in the final composite curve (see Supplementary data 1).

## Statistical methods – XRF

Core scanning XRF data[69] are shown in Figs. 2 and 6. To remove noise a locally-weighted scatterplot smoothing (LOWESS) method has been applied using the software PAST[67]. Samples with both % $CaCO_3$ and Ca count (XRF) data were cross plotted. A linear trendline was calculated for each site, and the linear equation used to create a secondary axis on figures, which translates Ca counts into approximate % $CaCO_3$ (Supplementary Fig. 13).

## Site information

IODP Site U1516 is located at a water depth of 2675 m, and U1513 at 2700 m – both lie in the Mentelle Basin, situated along the western margin of Australia (Supplementary Fig. 1). Holes U1516C and U1516D were drilled with a 20 m offset at 34° 20.9272′S, 112° 47.9711′E and 34° 20.9277′S, 112° 47.9573′E, respectively. Holes U1513A and U1513D were also drilled at with a 20 m offset, at 33°47.6084′S, 112°29.1338′E and 33°47.6196′S, 112°29.1339′E, respectively[36].

For this study, two successions of Cenomanian to Turonian chalk and clay-rich sediments were investigated from two spliced intervals of Holes U1516C and U1516D, and U1513A and U1513D. The splices are made using shipboard RGB colour data and Fe counts from core scanning XRF data to correlate overlapping sections of two holes drilled at the same site, providing a continuous interval unaffected by coring disturbances or poor recovery. All sample depths are reported in meters using the depth scale CCSF-M, which stands for Core Composite depth below Sea Floor, where Method "M" denotes that off-splice intervals are mapped to the splice[70].

IODP Sites U1516 and U1513 both transition from calcareous chalk interbedded with chert (< 470 m and < 234 m, respectively), grading down-section into greenish grey calcareous/nannofossil chalk with clay (Fig. 2). Below this are bands of black, light greenish grey, greenish grey, and very dark greenish grey claystone and clayey nannofossil chalk. Below 480 m (U1516), and 263 m (U1513), is a sequence of massive to mottled black and dark greenish grey nannofossil-rich claystone and claystone with nannofossils (Fig. 2).

A detailed sedimentological description of Site U1516, as well as details on drilling operations, logging, physical properties, magnetostratigraphy, and geochemistry are available in the IODP Expedition 369 proceedings report[36].

## Stratigraphy and site correlation

The biostratigraphic markers in Fig. 2 comprise a combination of initial IODP Expedition 369 shipboard biostratigraphy[36] and more recent publications[38,39] (Supplementary Tables 1 and 2). Within this framework, we propose a refined correlation between Sites U1516 and U1513 based on a combination of existing high resolution data (NGR and XRF) and high resolution data presented in this study ($^{13}C_{(TOC)}$, $^{13}C_{(carbonate)}$ and TOC).

Tie point A is placed between two peaks in NGR data. Due to core gaps and fewer geochemical data to support the correlation, it is more speculative. A slight excursion in the $^{13}C$ at Site U1516 and biostratigraphic constraints from *E. turriseiffellii* first occurrence (FO) (U1516) and *Ticinella primula* FO (U1513 – 306.88 – 304.91 m) suggest that this could represent a low resolution record of OAE 1d, which occurs during the latest Albian ~101 Ma[2] (Fig. 2). Although more isotope data is needed, our interpretation of OAE 1d coinciding with the clear elevation in NRG is consistent with the stratigraphically higher MCE and OAE2 events.

Tie point B is placed at the onset of a significant positive CIE in $^{13}C_{(organic)}$ (2‰ at U1513 and 1.5‰ at U1516) and $^{13}C_{(carbonate)}$ (1‰ at U1513 and 2‰ at U1516), and a concurrent spike in NGR data. Biostratigraphic markers indicate an age between 98 and 96 Ma, consistent

with the MCE (96.5 Ma)[8]. In other Northern Hemisphere records, the MCE is defined by a positive CIE ~1‰[71], and we here contribute a Southern Hemisphere record of the MCE. Tie point C is placed mid-way through the CIE at the second peak in NGR, and tie point D is placed at the end of the CIE. There is no significant increase in % wt. TOC across this interval, which is reflected in the lack of black shale horizons, however this is not uncommon in MCE sediments from other sites[7]. Tie point E is placed below *G. nanum* last occurrence (LO) when NGR falls to low values, and at the termination of a $^{13}C_{(carbonate)}$ negative CIE, after which stable values persist for > 5 m of sediment core.

Tie point F is placed at the occurrence of a short-lived negative shift in $^{13}C_{(carbonate)}$, modest NGR spike and Ca/Al fall. We interpret this as defining the base of OAE2, in agreement with biostratigraphic markers and recent publications[38,39] (Fig. 3). Tie points G and H are placed at the onset and termination of a low carbonate horizon, interpreted to occur within the OAE2 main phase. Specifically, tie point G is placed at the disappearance of carbonate, a spike in TOC content, elevated NGR and a negative excursion in $^{13}C_{(organic)}$. Tie point H is placed at the termination of the low carbonate interval when Ca/Al values increase from 0 and $^{13}C_{(carbonate)}$ data returns. The final tie point I is placed above *Q. gartneri* FO (near the end of OAE2), at maximum Ca/Al and where $^{13}C_{(carbonate)}$ data begins to decline after a long plateau phase.

## Reporting summary

Further information on research design is available in the Nature Portfolio Reporting Summary linked to this article.

## Data availability

The data generated in this study have been deposited in "figshare" under https://doi.org/10.6084/m9.figshare.25106123. Data generated in this study are also provided in Excel workbooks called Supplementary Data 1, 2, and 3. Benthic foraminifera are curated at the Camborne School of Mines, University of Exeter, Penryn Campus, Cornwall TR10 9EZ.

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

## Acknowledgements

We thank the IODP Expedition 369 Science Party, technicians, and crew for their hard work coring Sites U1513 and U1516, samples from which form the basis of this study. We are grateful to Hilary Sloane and Harvey Pickard for their support with stable isotope analyses, to Nicola Atkinson for her support with radiogenic isotope analyses, and to Thomas Gibson for his support with Hg analyses. This publication contains work conducted during a PhD study undertaken by C.A.W-T. as part of the Natural Environment Research Council (NERC) Doctoral Training Partnership (DTP) GW4+ (Grants NE/L002434/1 and NE/S007504/1). This work was supported by NERC National Environmental Isotope Facility (NEIF) Grants 2398.0421 and 2586.1022 (to S.K.), and NERC Isotope Geoscience Steering Committee (NIGFSC) Grant IP-1914-0619 (to S.K.). S.K., J.L., M.J.L., I.L.M., J.B.R., D.W. and C.A.W-T. publish with the approval of the Director, British Geological Survey (NERC).

## Author contributions

Study conceptualization by S.K., K.B., and C.A.W-T. Stable isotope analyses by C.A.W-T. and J.L., radiogenic isotope analyses by C.A.W-T., D.W. and I.L.M., Hg analyses by C.A.W-T. Micropalaeontological data by T.E. C.A.W-T., S.K., K.A.B., K.L., T.E., M.J.L., J.L., J.B.R., I.L.M., and D.W. contributed to data analysis, and to the drafting and editing of the manuscript.

## Competing interests

The authors declare no competing interests.
