## [Peer Review File · Nature Communications]

Oceanic Anoxic Event 2 triggered by Kerguelen Plateau volcanismREVIEWER COMMENTS

Reviewer #1 (Remarks to the Author):

M. R. Rampino, REVIEW OF: Oceanic Anoxic Event 2 triggered by Kerguelen Plateau volcanism Walker-Trivett, C.A., Kender, S., Bogus, K.A., Littler, K.L., Edvardsen, T., Leng, M.J., and Lacey, J.

Oceanic Anoxic Event 2 (OAE-2) occurred at the Cenomanian/Turonian boundary (~94 Ma) during hot-house climate conditions, characterized by high atmospheric pCO₂ produced by increased volcanism from ocean plateaus or contemporary CFB activity. The warming during OAE-2 apparently led to enhanced nutrient cycling and stimulated high productivity, explaining the deposition of organic-rich sediments, and the associated positive shift in δ¹³C seen during OAE-2. This mechanism may also have applied to the brief MCE anoxic interval.

All of the analytical work seems well done. Hg and Hg/TOC, Os-isotope ratios, and δ¹³C anomalies are all convincing. They add up to a volcanic source for the Hg, and a basaltic volcanism source for the Os-isotope anomaly, most probably from ocean plateau volcanism.

Numerous Hg/TOC pulses document early volcanic activity, and although this is not recorded in Os isotopes from the sites, precursor negative Os excursions do occur in other localities. But this precursor volcanism was likely relatively minor as it occurred prior to the major climate warming associated with OAE-2. It is clear that the largest Hg spike occurs at the beginning of the Osmium-isotope excursion and is associated with the initial negative δ¹³C pulse of OAE-2.

This paper highlights the difficulties of correlating ocean anoxic events with potential particular volcanic sources of the Hg and CO₂-driven climatic warming, either from Ocean Plateau volcanism or perhaps from CFB activity. The problem is that the Cretaceous LIPs have a wide range of dates including the Caribbean LIP ~95–83 Ma, HALIP ~130–90 Ma, and the Kerguelen Plateau ~122–90 Ma. So, although, each of these LIPs have major eruptive phases that could be compatible with OAE-2 and the MCE, the dating is not precise enough for convincing correlations between phases of volcanism and the anoxic intervals.

The elevated Hg and Hg/TOC in the Southern Hemisphere as opposed to Northern Hemisphere sites suggests the possibility of a volcanic source of the Hg on the Kerguelen Plateau. The Madagascar CFB, however, is dated at 92.9 ± 3.8 Ma, which is close to the age of OAE-2, and is also in the Southern Hemisphere.

Numerous Hg/TOC pulses document early volcanic activity, and although this is not recorded in Os isotopes in the study area, precursor negative Osmium-isotope excursions do occur in other localities. This precursor OAE-2 volcanism was likely relatively minor as it occurred before the major climate warming associated with OAE-2. The major Hg spike occurred at the beginning of the Os excursion, and is clearly associated with the initial negative δ¹³C anomaly.

But, with the problems in dating oceanic-plateau volcanic activity, and in establishing a close correlation, the authors can still only speculate on potential sources for the CO₂ warming and for the Hg anomalies associated with OAE-2. Thus, this is a solid paper on the association and timing of Hg and Hg/TOC anomalies, Os-isotope shifts, and δ¹³C anomalies related to OAE-2, and the analyses support a potential Southern Hemisphere volcanic source, but the authors still cannot pin down the Kerguelen Plateau as the specific source of the volcanism.

Reviewer #2 (Remarks to the Author):

Review of Walker-Trivett et al. "Oceanic Anoxic Event 2 triggered by Kerguelen Plateau volcanism" for Nature Communications

This study presents new geochemical proxy data from 2 well-described marine drill core sections from the Mentelle Basin (offshore southwestern Australia), that shed light on the environmental factors coinciding with 2 major Cretaceous oceanic anoxic episodes (OAE2 and the MCE). The authors synthesize global data from several studies to argue convincingly that Kerguelen Plateau volcanism was a primary trigger for global marine anoxia during OAE2 (~94 Ma), but not the MCE (~96.5 Ma). They make particular use of mercury and Hg/TOC (total organic carbon) abundances, a commonly used proxy for volcanic loading events (atmosphere, oceans and surface terrestrial environments) preserved in the stratigraphic record.

I find the paper to be extremely well-written, well-organized, and well-documented with data, figures and references. There are a few places where I feel the discussion could be expanded, clarified, or perhaps re-ordered for better clarity. The conclusions appear to be sound and well-argued, pointing the way to further hypothesis testing of the primary cause of these profoundly enigmatic global marine anoxia episodes in the geologic record.

A few minor comments/corrections:

Line 86 – specify 'marine hydrothermal emission'

Line 87 – specify that Hg is likely to remain in the 'ocean water column'

Line 88 – specify majority of 'marine Hg'

Line 89 – what form does Hg mainly take in marine sediments? Is it methyl Hg, or dominantly oxidized inorganic Hg (2+) bound to organic particulates? If the latter, please specify or elaborate on this here before introducing the Hg/TOC parameter in the next sentence.

Line 91 – '...is therefore surprising if...'

Line 93 – 'On the other hand, MCE initiation is understudied...'

Line 94 – '...and no consensus on MCE-related LIP involvement...'

Line 97 – 'located relatively close to the mid-to-late Cretaceous (~122-90 Ma) Kerguelen LIP.'

Line 97 – maybe specify a paleolatitude/paleodistance of Kerguelen Plateau vs. these sites?

Lines 107-109 – do you mean to specify volcanic emission from the Kerguelen LIP?

Lines 109-110 – is it 'little evidence' from the Hg/TOC trends in your dataset, or other studies?

Line 110 – are you inferring high productivity and possible upwelling for these sites only, or for the southern hemisphere record in general? For OAE2 only, or also the MCE? Please specify.

Line 111-118 – should specify here that both sites record histories across the OAE2 and MCE...

Line 152 – this discussion appears to cut off before any more details are given for the MCE at these sites. Is that intentional?

Line 159 – typo in the header "Kerguelan"

Line 159 – should the header be more specific, e.g. 'Sedimentary Hg reflects Kerguelen LIP activity at OAE2'?

Lines 172 – 177 – were sulfur concentrations obtained, or are there S data for these intervals?

Line 188 – 'high' refers to general background and not spikes, is that correct?

Line 190 – had better specify what OAE1d is...?

Line 194 – is this age overlapping within error of OAE2? If so, that should be stated.

Line 203 – do you mean OAE2, or is MCE included in this?

Lines 207-209 – not sure I understand this...do you mean the latitudinal position would have disproportionately influenced southern hemisphere sites? Please clarify.

Line 224 – expand header to read 'Evolution of Cenomanian volcanism from Hg and Os records'

Line 236 – probably should start this section summarizing the Hg and Hg/TOC for OAE2 in the Mentelle Basin, just as you did in the preceding paragraph for the MCE, before introducing the Os record for comparison.

Line 258 – '...LIP-related carbon release at OAE2.'

Line 384 – should this be an 'RA 915F LUMEX portable mercury analyzer'?

Line 409 – is this a typo, or do the authors mean that some Hg was below detection?

With these minor corrections/revisions, I believe this paper will be an important addition to the literature and of wide interest to the large community of scientists using past records of climate and environmental change to chart Earth's future. It is appropriate for Nature Communications.

Reviewer #3 (Remarks to the Author):

This manuscript examines evidence for the Oceanic Anoxic Event 2 (OAE2) being caused by a Large Igneous Province (LIP) eruption. This in itself is not a new conclusion as many papers have previously suggested a LIP driver for OAE2 based on various lines of evidence. The authors provide evidence of enhanced mercury concentrations from their section which they argue is more definitive proof of a volcanic driver. This is not in itself new either as Yao et al., 2022 recently published both mercury concentration and mercury isotope data and make the same conclusions (note here that the isotope data provides more definitive evidence than just mercury concentrations). This paper is not cited by the authors such that their discussion is out of context of the recent literature, and this makes the novelty of their findings less clear. While the literature debates the Caribbean LIP and/or High Arctic LIP as the driver of OAE2 the authors suggest here the Kerguelen LIP as an alternative. There are limited lines of argument in the manuscript though that support this is the case, or that the other LIP events are less likely. So as a general remark, I find the quality of the work, and presentation of results very well done. The discussion of the results though is lacking both current context and full argumentation to support the conclusions. This can be fixed easily I am sure, although I am not convinced that this will lead to novel findings as compared to more of an incremental discovery (this part will be more difficult). I am happy to be convinced otherwise and would thus recommend major revisions to see if the authors can address these concerns and pull out more novel conclusions.

Stephen Grasby

Other comments:

Ln 45: here maybe better to give the citations for each LIP directly rather than collectively in the next sentence on Ln 49.

Ln 49: I don't think citation 13 is ideal here, there is a more recent paper that updates volcanic ages and stratigraphy in the Sverdrup Basin (see Naber et al., 2020, GSA Bull).

Ln 60: Again, Naber et al. 2020 is a better citation here and it includes assessment of relative contributions of submarine and sub areal HALIP eruption along with linkage to OAE2. You can use both citations as well but Naber is the most recent.

Ln 80: Here and elsewhere (Ln 89) consider using Grasby et al. 2020 in Geology as a citation as it specifies models the global mercury cycle during a LIP event as compared to modern processes.

Ln 94: Might be good to include discussion in Percival as well as Naber on relative contributions of submarine and subareal volcanisms and Hg transport here. There is a bit late on but I think some background context at this point would be good.

Ln 159: This section title reads as a conclusion, so it seems a bit contrary to scientific writing style to start with your conclusion then provide the evidence.

Ln 159: I do not see any line of argument as to why the Kerguelan has to be the source of Hg, or why the other LIPs are not. If this is a main conclusion of the study than some element of discussion on

the topic is required. Otherwise it is just as fine to be neutral on the topic and discuss all the potential LIP events as a potential source. I am not sure there are enough data collectively (all published work) to really point a finger at any particular LIP at this point.

Ln 167: Here again, but this time more explicitly, the conclusion is given first prior to presentation and discussion of results. This seems counter to typical scientific writing style.

Ln 168: The Yao et al 2022 (and many other papers) provide mercury stable isotope data as a fingerprint to distinguish terrestrial versus atmospheric mercury input. So all the discussion here could simply be resolved by some isotope data and it would make this seem much less speculative.

Ln 232: While I applaud the authors for recognising the contradictory results of the Hg and Os records, this seems to then be quickly ignored. I think some discussion to explain why this contradiction exists is warranted. Note that the Yao et al. 2022 paper shows more consistency between Hg and Os records so this also needs discussion.

Ln 236: Unclear what is being referred to here by "Os records". What records? From where?

Ln 241: I am unclear what is meant by this sentence as it refers to the 'former' meaning the organic C isotope record at the start of the sentence but then ends stating that other records only use organic carbon records, so it seems like the same thing is being discussed but the wording implies that there are differences. The sentence needs to be restructured to make this point more clear.

Ln 278: the mention on isotopes decreasing needs clarification as many models would argue increased carbon burial would lead to increased carbon isotope values due to sequestration of light carbon.

We would like to thank our three reviewers for taking the time to make insightful suggestions and point out key issues, which we hope we have now fully addressed. Here we outline our point-by-point response to each reviewer comment, explaining how we have now addressed it now and where we have modified the manuscript. We think we have addressed all the points and very much welcome further feedback if required.

In summary, in addition to the numerous detailed changes made, the main change is the addition of two new datasets sedimentary Nd and Sr isotopes (please see responses below regarding why). We have also therefore included the new data, full methods and new co-authors.

Reviewer #1 (Remarks to the Author):

M. R. Rampino, REVIEW OF: Oceanic Anoxic Event 2 triggered by Kerguelen Plateau volcanism
Walker-Trivett, C.A., Kender, S., Bogus, K.A., Littler, K.L., Edvardsen, T., Leng, M.J.,
and Lacey, J.

Oceanic Anoxic Event 2 (OAE-2) occurred at the Cenomanian/Turonian boundary (~94 Ma) during hot-house climate conditions, characterized by high atmospheric pCO₂ produced by increased volcanism from ocean plateaus or contemporary CFB activity. The warming during OAE-2 apparently led to enhanced nutrient cycling and stimulated high productivity, explaining the deposition of organic-rich sediments, and the associated positive shift in $\delta^{13}\text{C}$ seen during OAE-2. This mechanism may also have applied to the brief MCE anoxic interval.

All of the analytical work seems well done. Hg and Hg/TOC, Os-isotope ratios, and $\delta^{13}\text{C}$ anomalies are all convincing. They add up to a volcanic source for the Hg, and a basaltic volcanism source for the Os-isotope anomaly, most probably from ocean plateau volcanism.

Numerous Hg/TOC pulses document early volcanic activity, and although this is not recorded in Os isotopes from the sites, precursor negative Os excursions do occur in other localities. But this precursor volcanism was likely relatively minor as it occurred prior to the major climate warming associated with OAE-2. It is clear that the largest Hg spike occurs at the beginning of the Osmium-isotope excursion and is associated with the initial negative $\delta^{13}\text{C}$ pulse of OAE-2.

This paper highlights the difficulties of correlating ocean anoxic events with potential particular volcanic sources of the Hg and CO₂-driven climatic warming, either from Ocean Plateau volcanism or perhaps from CFB activity. The problem is that the Cretaceous LIPs have a wide range of dates including the Caribbean LIP ~95–83 Ma, HALIP ~130–90 Ma, and the Kerguelen Plateau ~122–90 Ma. So, although, each of these LIPs have major eruptive phases that could be compatible with OAE-2 and the MCE, the dating is not precise enough for convincing correlations between phases of volcanism and the anoxic intervals.

The elevated Hg and Hg/TOC in the Southern Hemisphere as opposed to Northern Hemisphere sites suggests the possibility of a volcanic source of the Hg on the Kerguelen Plateau. The Madagascar

CFB, however, is dated at 92.9 ± 3.8 Ma, which is close to the age of OAE-2, and is also in the Southern Hemisphere.

This is clearly a critical point to the study, and it is also raised by reviewer 3. Whilst the palaeogeography of ~ 94 Ma demonstrates poor connectivity between the Madagascan LIP (MLIP) and the Mentelle Basin (Fig. 1), such that we suggest the raised Hg is more likely to have come from a more proximal sources, we concede the timing of the event and potential atmospheric Hg loading does not preclude the MLIP in itself. We have therefore generated and presented a new Nd and Sr isotope dataset (and included 3 new co-authors), which indicates that an increasing proportion of sediments in the Mentelle Basin were being sourced from radiogenic younger volcanic rocks throughout the Cenomanian, between the MCE and OAE2 (new data shown on Fig. 2; also see new Fig. 6 and Supplementary Figs 6-8). The data strongly suggest uplift and erosion of a nearby radiogenic source was occurring in the lead up to OAE2, further supporting enhanced Kerguelen LIP activity as the cause of OAE2. We have added the following text:

“To test for changes in sediment provenance consistent with an uplifting landmass from nearby LIP activity, we build on a recent study of Site U1516⁴² and measure fine fraction sedimentary ϵ Nd and $^{87}\text{Sr}/^{86}\text{Sr}$ at U1513 (Fig. 2). The $<2 \mu\text{m}$ fraction values are considered consistent between craton sources and ocean basins⁵⁰, and we find similar values in coarser fractions supporting this provenance proxy (Supplementary Fig. 6). A long-term shift to higher (more radiogenic) ϵ Nd and lower $^{87}\text{Sr}/^{86}\text{Sr}$ values occurs in the build-up to (between tie points C and G) and over OAE2, indicative of an increase in the proportion of sedimentary material deposited in the Mentelle Basin from erosion of younger rocks such as the Bunbury Basalts (~ 130 - 137 Ma),⁵¹ Naturaliste Plateau (Early Cretaceous)⁵² or Kerguelen Plateau (Fig. 6, Supplementary Fig. 7). In contrast, less radiogenic values occur during the MCE and the low carbonate interval of OAE2 (also measured in U1516)⁴² which have been interpreted as resulting from an increased hydrological cycle reactivating continental river systems and eroding the Yilgarn Craton⁴². Given the dramatic sedimentation rate increase of U1513 after the onset of elevated Hg/TOC at ~ 260 mbsf (averages of 0.3 cm kyr^{-1} below, to 2.2 cm kyr^{-1} above, Supplementary Fig. 8), this extra detrital input must have come from a newly available radiogenic continental source, rather than simply the result of a reduced hydrological cycle and a (necessarily smaller) input of more local Bunbury Basalts. Whether this was from a newly uplifted Kerguelen Plateau and/or Naturaliste Plateau – the former we suggest as more likely – our data support concurrent Kerguelen LIP activity (rather than the far-field Madagascar LIP; Fig. 1) as the source of the coeval Hg pulses and local increase in radiogenic sedimentation.” Lines 220-240

Numerous Hg/TOC pulses document early volcanic activity, and although this is not recorded in Os isotopes in the study area, precursor negative Osmium-isotope excursions do occur in other localities. This precursor OAE-2 volcanism was likely relatively minor as it occurred before the major climate warming associated with OAE-2. The major Hg spike occurred at the beginning of the Os excursion, and is clearly associated with the initial negative $\delta^{13}\text{C}$ anomaly.

But, with the problems in dating oceanic-plateau volcanic activity, and in establishing a close correlation, the authors can still only speculate on potential sources for the CO₂ warming and for the Hg anomalies associated with OAE-2. Thus, this is a solid paper on the association and timing of Hg and Hg/TOC anomalies, Os-isotope shifts, and $\delta^{13}\text{C}$ anomalies related to OAE-2, and the analyses support a potential Southern Hemisphere volcanic source, but the authors still cannot pin down the

Kerguelen Plateau as the specific source of the volcanism.

We now feel any previous speculation in the manuscript is removed; specifically, correlatable Hg pulses in the Mentelle Basin evidence volcanism from the Southern Hemisphere as the trigger of OAE2, and a shift towards increased sedimentation of radiogenic material fingerprints activity of the Kerguelen LIP.

Reviewer #2 (Remarks to the Author):

Review of Walker-Trivett et al. "Oceanic Anoxic Event 2 triggered by Kerguelen Plateau volcanism" for Nature Communications

This study presents new geochemical proxy data from 2 well-described marine drill core sections from the Mentelle Basin (offshore southwestern Australia), that shed light on the environmental factors coinciding with 2 major Cretaceous oceanic anoxic episodes (OAE2 and the MCE). The authors synthesize global data from several studies to argue convincingly that Kerguelen Plateau volcanism was a primary trigger for global marine anoxia during OAE2 (~94 Ma), but not the MCE (~96.5 Ma). They make particular use of mercury and Hg/TOC (total organic carbon) abundances, a commonly used proxy for volcanic loading events (atmosphere, oceans and surface terrestrial environments) preserved in the stratigraphic record.

I find the paper to be extremely well-written, well-organized, and well-documented with data, figures and references. There are a few places where I feel the discussion could be expanded, clarified, or perhaps re-ordered for better clarity. The conclusions appear to be sound and well-argued, pointing the way to further hypothesis testing of the primary cause of these profoundly enigmatic global marine anoxia episodes in the geologic record.

A few minor comments/corrections:

Line 86 – specify 'marine hydrothermal emission'

This has now been changed.

Line 87 – specify that Hg is likely to remain in the 'ocean water column'

This has been changed.

Line 88 – specify majority of 'marine Hg'

This is changed.

Line 89 – what form does Hg mainly take in marine sediments? Is it methyl Hg, or dominantly oxidized inorganic Hg (2+) bound to organic particulates? If the latter, please specify or elaborate on this here before introducing the Hg/TOC parameter in the next sentence.

This has now been addressed in lines 86-88.

Line 91 – '...is therefore surprising if...'

This has been changed.

Line 93 – ‘On the other hand, MCE initiation is understudied...’

This has been changed.

Line 94 – ‘...and no consensus on MCE-related LIP involvement...’

This has been changed.

Line 97 – ‘located relatively close to the mid-to-late Cretaceous (~122-90 Ma) Kerguelen LIP.’

This has been changed.

Line 97 – maybe specify a paleolatitude/paleodistance of Kerguelen Plateau vs. these sites?

We have identified the approximate palaeolatitude of the Kerguelen Plateau in line 101, and refer readers to Figure 4, which demonstrates the proximity of the Mentelle Basin to the Kerguelen LIP.

“Here, we measure Hg and Hg/TOC at two high palaeolatitude (~60°S) sites in the Mentelle Basin⁴¹ – located relatively close to the mid-to-late Cretaceous (~122–90 Ma) Kerguelen LIP (Fig. 1, spanning ~50-62°S)”

Lines 107-109 – do you mean to specify volcanic emission from the Kerguelen LIP?

We hoped to identify any evidence of aerial volcanic emissions across this time span (which would be present at other sites around the world), or evidence of a nearby source if subaerial, indicating potentially a southern hemisphere LIP, likely Kerguelen.

Lines 109-110 – is it ‘little evidence’ from the Hg/TOC trends in your dataset, or other studies?

This has been addressed by lines 116-117.

Line 110 – are you inferring high productivity and possible upwelling for these sites only, or for the southern hemisphere record in general? For OAE2 only, or also the MCE? Please specify.

This has been addressed by lines 117-118.

Line 111-118 – should specify here that both sites record histories across the OAE2 and MCE...

This has been addressed in line 126.

Line 152 – this discussion appears to cut off before any more details are given for the MCE at these sites. Is that intentional?

The MCE is identified by a positive C isotope excursion, as expected, and so requires less investigation than OAE2, which lacks the characteristic positive excursion associated with most sites due to the low carbonate zone.

Line 159 – typo in the header “Kerguelan”

This has been changed.

Line 159 – should the header be more specific, e.g. ‘Sedimentary Hg reflects Kerguelen LIP activity at OAE2’?

This has been changed.

Lines 172 – 177 – were sulfur concentrations obtained, or are there S data for these intervals?

S counts are available through these intervals (core-scanning XRF data), but are not ground-truthed with S concentration data. Sulfur / Aluminium counts are displayed in an additional supplementary figure.

Line 188 – ‘high’ refers to general background and not spikes, is that correct?

Actually, it does refer to the spikes – clarified in line 205.

Line 190 – had better specify what OAE1d is...?

This has been addressed by lines 207-208.

Line 194 – is this age overlapping within error of OAE2? If so, that should be stated.

This has been addressed in line 214

Line 203 – do you mean OAE2, or is MCE included in this?

This has been addressed in line 251. OAE 2 only.

Lines 207-209 – not sure I understand this...do you mean the latitudinal position would have disproportionately influenced southern hemisphere sites? Please clarify.

This has been addressed by lines 257-260.

Line 224 – expand header to read ‘Evolution of Cenomanian volcanism from Hg and Os records’

This has been changed.

Line 236 – probably should start this section summarizing the Hg and Hg/TOC for OAE2 in the Mentelle Basin, just as you did in the preceding paragraph for the MCE, before introducing the Os record for comparison.

This has been addressed by lines 284-288.

Line 258 – ‘...LIP-related carbon release at OAE2.’

This has been changed.

Line 384 – should this be an ‘RA 915F LUMEX portable mercury analyzer’?

This has been changed.

Line 409 – is this a typo, or do the authors mean that some Hg was below detection?

This has been addressed by lines 462-463.

With these minor corrections/revisions, I believe this paper will be an important addition to the literature and of wide interest to the large community of scientists using past records of climate and environmental change to chart Earth’s future. It is appropriate for Nature Communications.

Reviewer #3 (Remarks to the Author):

This manuscript examines evidence for the Oceanic Anoxic Event 2 (OAE2) being caused by a Large Igneous Province (LIP) eruption. This in itself is not a new conclusion as many papers have previously suggested a LIP driver for OAE2 based on various lines of evidence. The authors provide evidence of enhanced mercury concentrations from their section which they argue is more definitive proof of a volcanic driver. This is not in itself new either as Yao et al., 2022 recently published both mercury concentration and mercury isotope data and make the same conclusions (note here that the isotope data provides more definitive evidence than just mercury concentrations). This paper is not cited by the authors such that their discussion is out of context of the recent literature, and this makes the novelty of their findings less clear. While the literature debates the Caribbean LIP and/or High Arctic LIP as the driver of OAE2 the authors suggest here the Kerguelen LIP as an alternative. There are limited lines of argument in the manuscript though that support this is the case, or that the other LIP events are less likely.

We would like to draw attention to our new Nd and Sr isotope dataset for sedimentary provenance, in our detailed response to reviewer 1 (above). This provides a very strong second line of evidence that, along with the Hg spikes in this section, fingerprints Kerguelen LIP as the trigger for OAE2.

So as a general remark, I find the quality of the work, and presentation of results very well done. The discussion of the results though is lacking both current context and full argumentation to support the conclusions. This can be fixed easily I am sure, although I am not convinced that this will lead to novel findings as compared to more of an incremental discovery (this part will be more difficult). I am happy to be convinced otherwise and would thus recommend major revisions to see if the authors can address these concerns and pull out more novel conclusions.

We are now in a position, with the new Nd and Sr isotope dataset, to argue strongly for a novel finding with this paper that should be considered far more than incremental. We are the first to provide strong and convincing evidence that Kerguelen LIP activity was responsible for triggering OAE2.

Stephen Grasby

Other comments:

Ln 45: here maybe better to give the citations for each LIP directly rather than collectively in the next sentence on Ln 49.

We found that introducing this appeared to interrupt the flow of the argument, but will put this in if you still think it is necessary.

Ln 49: I don't think citation 13 is ideal here, there is a more recent paper that updates volcanic ages and stratigraphy in the Sverdrup Basin (see Naber et al., 2020, GSA Bull).

This has been changed.

Ln 60: Again, Naber et al. 2020 is a better citation here and it includes assessment of relative contributions of submarine and sub areal HALIP eruption along with linkage to OAE2. You can use both citations as well but Naber is the most recent.

This has been changed.

Ln 80: Here and elsewhere (Ln 89) consider using Grasby et al. 2020 in Geology as a citation as it specifies models of the global mercury cycle during a LIP event as compared to modern processes.

This has been changed.

Ln 94: Might be good to include discussion in Percival as well as Naber on relative contributions of submarine and subareal volcanisms and Hg transport here. There is a bit late on but I think some background context at this point would be good.

This has now been addressed in lines 92-96.

Ln 159: This section title reads as a conclusion, so it seems a bit contrary to scientific writing style to start with your conclusion then provide the evidence.

Upon reflection we do think it likely to be a useful guide to anyone reading through the paper to understand the main point of the preceding text, but are happy to amend if you think necessary.

Ln 159: I do not see any line of argument as to why the Kerguelan has to be the source of Hg, or why the other LIPs are not. If this is a main conclusion of the study then some element of discussion on the topic is required. Otherwise it is just as fine to be neutral on the topic and discuss all the potential LIP events as a potential source. I am not sure there are enough data collectively (all published work) to really point a finger at any particular LIP at this point.

We very much agree with this point, and have added text discussing the localised signal of Hg at the Mentelle Basin – which is significantly higher than at any other OAE2 site – and how ocean currents and paleogeography likely leave Kerguelen as the reasonable source. The Madagascan and Caribbean LIPs, and the HALIP are all isolated from the Mentelle Basin by distance, continental arrangement, and restricted shallow ocean passages. The lack of consistent global signal indicates that a largely submarine eruption is likely, and thus ocean transport the most likely system of spreading the Hg signal. But we do concede that this line of reasoning does not completely exclude the Madagascan LIP which theoretically could have injected atmospheric Hg but that was not carried across the equator to Gulf of Mexico/Arctic regions.

Therefore, we add a further dataset of Nd and Sr isotope data for sediment provenance, which demonstrate an increasing proportion of sediment in the Mentelle Basin coming from a newly uplifted nearby Plateau from the mid-Cenomanian times (please see detailed description above). Critically, the sediment products in the Mentelle basin cannot have come from erosion of far field plateaus such as MLIP. Furthermore, these data correlate with the increased Hg spikes seen at Site U1513, further suggesting a connection between Kerguelen LIP volcanic activity and the Hg concentrations and Hg/TOC in the Mentelle Basin.

Ln 167: Here again, but this time more explicitly, the conclusion is given first prior to presentation and discussion of results. This seems counter to typical scientific writing style.

As above, we do think on balance it is likely to be a useful guide to anyone reading through the paper, but are happy to amend if you think necessary.

Ln 168: The Yao et al 2022 (and many other papers) provide mercury stable isotope data as a fingerprint to distinguish terrestrial versus atmospheric mercury input. So all the discussion here could simply be resolved by some isotope data and it would make this seem much less speculative.

Unfortunately, it is not within the scope of our research budget or laboratories to carry out this kind of analysis, but feel with the totality of our datasets we have a compelling interpretation. We will explore future possibility of mercury isotopes if there is sufficient material for this.

Ln 232: While I applaud the authors for recognising the contradictory results of the Hg and Os records, this seems to then be quickly ignored. I think some discussion to explain why this contradiction exists is warranted. Note that the Yao et al. 2022 paper shows more consistency between Hg and Os records so this also needs discussion.

We agree with this point and now discuss more fully that Os and Hg are proxies for very different process related to LIP activity, and act on different timescales and geographic extent. We have added the following to lines 284-295:

“It is therefore possible that submarine LIP activity may demonstrate a global $Os_{(i)}$ excursion, whilst Hg is more restricted in its submarine dispersal, showing only local/regional excursions. Thus, any submarine LIP activity, not just the Kerguelen Plateau, may be influencing $Os_{(i)}$ data globally. Conversely, the Hg record would be restricted to nearby strata, necessitating a proximal source.”

Ln 236: Unclear what is being referred to here by “Os records”. What records? From where?

This has been addressed by lines 288-290, where we refer to which records:

“Os records over OAE2 from the Maverick Basin (WIS)²², Tibet⁶, and the Vocontian Trough, south-east France²⁶,”

Ln 241: I am unclear what is meant by this sentence as it refers to the ‘former’ meaning the organic C isotope record at the start of the sentence but then ends stating that other records only use organic carbon records, so it seems like the same thing is being discussed but the wording implies that there are differences. The sentence needs to be restructured to make this point more clear.

This has been addressed by lines 300-302 – amending a typo that referred to the organic fraction twice:

“ $^{13}C_{(carbonate)}$ records frequently exhibit a delayed or absent positive CIE (e.g., Mentelle Basin, Tibet, Vocontian Basin; Fig. 4), whilst important sites from the WIS are based solely on $\delta^{13}C_{(organic)}$.”

Ln 278: the mention on isotopes decreasing needs clarification as many models would argue increased carbon burial would lead to increased carbon isotope values due to sequestration of light carbon.

We absolutely agree, many thanks. This has been addressed in line 339.

REVIEWERS' COMMENTS

Reviewer #2 (Remarks to the Author):

The authors have produced superior manuscript in revised form that achieves the standards required for publication, in my opinion. The addition of radiogenic Nd-Sr isotopic data for the study site provides a welcome added dimension, helping to bolster their arguments. I have very few comments on the current version:

- 1) Line 164 shouldn't the subtitle to this section refer to Nd-Sr isotopes (not just Nd, Sr)?
- 2) Line 241 What is 'radiogenic sedimentation'?
- 3) Line 361 Perhaps the final, summary paragraph could be expanded by a line or two, referring back to the abstract's conclusions re: a refined geochemical signature for Kerguelen LIP volcanism?

Detailed figures and captions appear to be accurate as well as central to the presentation. I have no further comments except to commend the authors on a job well done, and to recommend publication.

Reviewer #3 (Remarks to the Author):

In general the authors have satisfactorily addressed all of my concerns in this revision. They have an interesting and relevant set of high quality data which will inform further the debate on OAE2. I suspect many may question the main conclusion as given in the title, however. I am not certain that the authors provide definitive proof that OAE 2 was caused just by the Kerguelen event. There was extensive volcanism around the world at the time and it could very well be a combination of eruptions, or the Kerguelen was the proverbial straw that broke the camels back. It might be fruitful to recognise this aspect towards the end of the paper. I believe though that the work does provide significant insight into the importance of the Kerguelen eruption and will be of broad interest. As one general comment, even with my brand new glasses I found the micro-text in many of the figures near impossible to read. They will need revision or reformatting. Otherwise the preparation of the text is excellent, it is a very well written manuscript.

Stephen Grasby

We would like to thank our reviewers for again taking the time to point out issues in our manuscript, which I hope we have now addressed fully. Here we outline our point-by-point response to each reviewer comment, explaining how we have now addressed it and where we have modified the manuscript.

Reviewer #2 (Remarks to the Author):

The authors have produced superior manuscript in revised form that achieves the standards required for publication, in my opinion. The addition of radiogenic Nd-Sr isotopic data for the study site provides a welcome added dimension, helping to bolster their arguments. I have very few comments on the current version:

- 1) Line 164 shouldn't the subtitle to this section refer to Nd-Sr isotopes (not just Nd, Sr)?

Yes - this has been changed.

- 2) Line 241 What is 'radiogenic sedimentation'?

We have expanded lines 241-242 to "local increase in the deposition of radiogenic sediments."

- 3) Line 361 Perhaps the final, summary paragraph could be expanded by a line or two, referring back to the abstract's conclusions re: a refined geochemical signature for Kerguelen LIP volcanism?

Lines 371-372 have been added to refer our findings back to the abstract's conclusions.

Detailed figures and captions appear to be accurate as well as central to the presentation. I have no further comments except to commend the authors on a job well done, and to recommend publication.

Reviewer #3 (Remarks to the Author):

In general the authors have satisfactorily addressed all of my concerns in this revision. They have an interesting and relevant set of high quality data which will inform further the debate on OAE2. I suspect many may question the main conclusion as given in the title, however. I am not certain that the authors provide definitive proof that OAE 2 was caused just by the Kerguelen event. There was extensive volcanism around the world at the time and it could very well be a combination of eruptions, or the Kerguelen was the proverbial straw that broke the camels back. It might be fruitful to recognise this aspect towards the end of the paper.

We have added lines 271-277, to address this comment and further explain why we attribute the enhanced volcanism signal to Kerguelen, rather than other active LIPs at the time.

I believe though that the work does provide significant insight into the importance of the Kerguelen eruption and will be of broad interest. As one general comment, even with my brand new glasses I found the micro-text in many of the figures near impossible to read. They will need revision or reformatting.

Text size has been significantly increased to ensure readability.

Otherwise the preparation of the text is excellent, it is a very well written manuscript.

Stephen Grasby